Risk factors for recovery from oculomotor nerve palsy after aneurysm surgery: a meta-analysis

Li Yuan 1
Zhao Ming 1
Li Xuemei 1
Liu Tiejuan 1
Zheng Lin 1
Hu Deyu 1
Liu Tongyan 2
Zhou Lingyun 1 no1zhly@163.com
1 The First Affiliated Hospital of Harbin Medical University , Harbin , China
2 Heilongjiang University of Chinese Medicine , Heilongjiang , China
Pedrino Gustavo
Electronic publication date: 2024 Oct 29
Publication date: 2024
Volume: 12
Electronic Location ID: e18207
Received 2024 Mar 8; Accepted 2024 Sep 10
Copyright: © 2024 Li et al.
Copyright year: 2024
Copyright holder: Li et al.
License: This is an open access article distributed under the terms of the Creative Commons Attribution License, which permits unrestricted use, distribution, reproduction and adaptation in any medium and for any purpose provided that it is properly attributed. For attribution, the original author(s), title, publication source (PeerJ) and either DOI or URL of the article must be cited.
License URL: https://creativecommons.org/licenses/by/4.0/

Keywords: Oculomotor nerve palsy, Aneurysm, Risk factors

Funding: The authors received no funding for this work.

==============================
Background

Risk factors for recovery from oculomotor nerve palsy (ONP) after aneurysm surgery explored by meta-analysis.

Methods

The PubMed, Embase, web of science, Cochrane library, China Knowledge, Wan fang, and VIP databases were searched for case-control or cohort studies on risk factors of oculomotor nerve palsy recovery after aneurysm surgery, with a cut-off date of 14 February 2024, and data were analyzed using Stata 15.

Result

A total of 12 articles involving 866 individuals were included, meta-analysis results suggesting that gender (OR = 0.75, 95% CI [0.51–1.10]), age (OR = 1.00, 95% CI [0.93–1.07]), aneurysm size (OR = 0.85, 95% CI [−0.71 to 1.01]), treatment time (OR = 1.01, 95% CI [0.91–1.13]) is not a risk factor for recovery of motor nerve palsy after aneurysm surgery. Preoperative complete ONP (OR = 2.27, 95% CI [1.07–4.81]), surgery (OR = 9.88, 95% CI [2.53–38.57]), subarachnoid hemorrhage (OR = 1.29, 95% CI [1.06–1.56]) is a risk factor for recovery of motor nerve palsy after aneurysm surgery.

Conclusion

Based on the results of the studies we included, we found that complete ONP before surgery led to poorer recovery, but patients with post-operative and subarachnoid hemorrhage had better recovery.

Introduction

Oculomotor nerve palsy (ONP) is a neurological syndrome that can be caused by a variety of etiological factors leading to oculomotor nerve involvement (Kaeser & Brodsky, 2013; Douglas et al., 2022), with the main manifestations of ptosis, limited eye movement, pupil dilatation, blurred or absent reaction to light, and double vision (Wu et al., 2023; Omodaka et al., 2022). The oculomotor nerve is the third pair of cerebral nerves, originating from the midbrain, exiting the brain through the inter-foot fossa, traveling to the upper part of the lateral wall of the cavernous sinus, and dividing into two branches from the supraorbital fissure into the orbit (Hong et al., 2023), which mainly innervates the motor nerve fibers of the rectus superioris, rectus inferiors, rectus medialis, trapezius inferiors, and levator palpebralis muscles as well as the autonomic fibers of the pupillary sphincter and the ciliary muscles (Yusufoğlu, Kobat & Keser, 2022; Kawakami et al., 2020), and is usually divided into the brainstem segment, the cavernous sinus segment, the orbital segment, and the cerebro-orbital segment in the clinic (Zu et al., 2017; Suda, Matsushita & Minamide, 2022). Any lesion from the brainstem to the orbit can cause complete or incomplete paralysis of the OMNP. The etiology of ONP is complex, and intracranial aneurysm (IA) is one of the most dangerous and common causes of ONP (Kameda-Smith et al., 2022; Jeeva-Patel, Mandell & Margolin, 2021).

IA is a cystic dilatation of the intracranial artery wall caused by a variety of pathologic factors (Saito et al., 2021). Intracranial aneurysms often occur at the bifurcation of the arteries that form the ring of Willis, where the vessel wall is most vulnerable to hemodynamic influences (Hall et al., 2017). Posterior communicating artery aneurysm (PcomAA) accounts for approximately 45.9% of all intracranial aneurysms and is the most common type of intracranial aneurysm. Subarachnoid hemorrhage (SAH) and ONP are common complications of PcomAA, and because of their anatomical proximity, recent reports have shown that 29.8% of ONPs are caused by IA (Bizilis, Simonin & Lind, 2021; Bertulli, Reinert & Robert, 2018). Current craniotomy and interventional embolization are both effective treatments for intracranial aneurysms. In previous studies on PcomAA causing ONP, some investigators favored cranial clamping as the best treatment (Vaphiades & Roberson, 2017). Because of the high incidence and poor prognosis of ONP, the need to determine which factors affect the prognosis of ONP is a more critical topic in clinical practice, but there is still a controversy about the prognostic factors in clinical practice, this study fills the gaps of individual studies with small sample sizes, inconsistent results, and difficulty in quantifying the impact of risk factors. Through comprehensive analysis, this study identified the key risk factors affecting the recovery of oculomotor nerve palsy, provided more statistical power and clinically guiding conclusions, and provided more reliable evidence support for clinical decision making.

Materials and Methods

This scheme is based on the preferred reporting items of the Protocol for Systematic Reviews and Meta-analyses (PRISMA-P). The review will be conducted according to PRISMA criteria (Liberati et al., 2009). The registration number was CRD42024508137.

Literature search

Two authors (LY and ZM) searched the CNKI, Wan Fang, VIP, PubMed, Embase, Cochrane Library, Web of Science, and other databases on case-control or cohort studies on risk factors of oculomotor nerve palsy recovery after aneurysm surgery. Searches were conducted between the start of database establishment and 14 February 2024. Subjects and free words were used for retrieval: intracranial aneurysm, oculomotor nerve diseases, risk factors. See Material S1 for the specific retrieval strategies.

Inclusion and exclusion criteria

Inclusion criteria were adults with diagnostic criteria for aneurysms, and exposure factors were case-control studies or cohort studies of postoperative motor nerve palsy. The primary outcome measure was multivariate risk factor analysis.

Exclusion criteria were meeting abstracts, meta-analyses, protocols, letters, duplicate publications, systematic reviews, failure to obtain full text, failure to obtain available data, and animal experiments.

Data extraction

Two independent evaluators (LY and ZM) independently screened the literature to extract the data and directly screened the easily judged literature by reading the titles and abstracts of the literature and the full text; for any disagreements, they consulted relevant experts (ZLY). The inclusion and exclusion criteria were strictly followed during the screening process. They extracted the corresponding indicators from the studies and crosschecked the extracted data to ensure consistency. The main data extracted included the name of the first author, year of publication, country, study design, sample size, gender, mean age, and tumor type.

Quality evaluation

The Newcastle-Ottawa Scale (NOS) (Stang, 2010) was used to evaluate case-control studies, including the selection of the study population (four points), comparability between groups (two points), and measurement of exposure factors or results (three points). The total score of the scale is nine, with ≤4 indicating low quality, 5–6 indicating medium quality, and ≥7 indicating high quality. If the two researchers disagree on the evaluation process, they will discuss the decision or ask a third party to decide.

Statistical analysis

Stata 15.0 was used to statistically analyze the data. The risk values for each study were described using the RR values and 95% confidence intervals (CIs) were calculated. The heterogeneity test (Q test) and I2 statistics were used to select the appropriate model for calculating the pooled RR. If I2 was greater than 50%, the random effects model was adopted; if I2 was less than or equal to 50%, the fixed effects model was adopted. For I2 > 50%, we assessed the sensitivity of the literature using the leave-one-out method. Additionally, we conducted a publication bias using the Egger test, with significance level set at α = 0.05. A P-value < 0.05 was considered statistically significant.

Results

Literature retrieval process

By searching PubMed, Embase, Web of Science, the Cochrane library, China Knowledge, Wan fang and VIP databases, a total of 173 papers were retrieved after removing the literature by reading the title, abstract and full text. Twelve literatures were finally included. The literature search flowchart is shown in Fig. 1.

Figure 1 Flow chart of prisma literature search.

Basic characteristics of literature

A total of 12 articles (Engelhardt et al., 2015; Gao et al., 2017; Hu et al., 2021; Liu et al., 2020; Chang et al., 2023; Wang et al., 2015; Xin et al., 2020; Qing et al., 2019; Song, Yin & Wang, 2012; Tan et al., 2014; Zhong et al., 2019; Wu et al., 2020) involving 866 individuals were included, four articles (Engelhardt et al., 2015; Gao et al., 2017; Zhong et al., 2019; Wu et al., 2020) were cohort studies, and eight articles (Hu et al., 2021; Liu et al., 2020; Chang et al., 2023; Wang et al., 2015; Xin et al., 2020; Qing et al., 2019; Song, Yin & Wang, 2012; Tan et al., 2014) were case-controls, and the basic characteristics of the literature are shown in Table 1. A total of 12 articles were included, and the NOS quality rating: two articles scored six, with moderate quality of studies. The rest all scored 7~8, and the overall quality of the included studies improved. The specific quality assessment is shown in Table 2.

Table 1 Basic characteristics of the literature.

Study	Year	Study design	Country	Sample size	Gender (M/F)	Mean age	Tumor type	
Engelhardt	2015	Case control	France	23	4/19	47.8	Internal aneurysm	
Gao	2017	Case control	China	52	8/44	53.9	Intracranial aneurysm	
Hu	2021	Cohort study	China	128	60/68	52.8	Intracranial aneurysm	
Liu	2020	Cohort study	China	152	70/82	55.6	Posterior communicating aneurysm	
Zhong	2019	Case control	China	102	85/17	59.8	Posterior communicating aneurysm	
Liu	2023	Cohort study	China	68	8/60	56.59	Posterior communicating aneurysm	
Wu	2020	Case control	China	19	1/18	53	Posterior communicating aneurysm	
Song	2012	Cohort study	China	31	8/23	50.4	Posterior communicating aneurysm	
Wang	2015	Cohort study	China	55	16/39	52	Posterior communicating aneurysm	
Xue	2020	Cohort study	China	50	8/42	57	Posterior communicating aneurysm	
Tan	2014	Cohort study	China	132	58/74	55	Posterior communicating aneurysm	
Zhao	2019	Cohort study	China	54	NR	55	Posterior communicating aneurysm	

Table 2 NOS score.

Case control	
Study	Is the case definition adequate?	Representativeness of the cases	Determination of control group	Definition of controls	Comparability of cases and controls based on the design or analysis	Ascertainment of exposure	Same method of ascertainment for cases and controls	Non response	Total scores	
Engelhardt et al. (2015)	*	*	*	*	**	*	*	*	8	
Gao et al. (2017)	*	*	*	*	*	*	*	*	7	
Zhong et al. (2019)	*	*	*	*	*	*	*	*	7	
Wu et al. (2023)	*	*	*	*	**	*	*	*	8	
Cohort study	
Study	Representativeness of the exposed group	Selection of non-exposed groups	Determination of exposure factors	Identification of outcome indicators not yet to be observed at study entry	Comparability of exposed and unexposed groups considered in design and statistical analysis	Design and statistical analysis	Adequacy of the study’s evaluation of the outcome	Adequacy of follow-up in exposed and unexposed groups	Total scores	
Hu et al. (2021)	*	*	*	*	**	*	*	*	8	
Liu et al. (2020)	*	*	/	*	*	*	*	*	6	
Chang et al. (2023)	*	*	*	*	*	*	*	*	7	
Song, Yin & Wang (2012)	*	*	*	*	*	*	*	*	7	
Wang et al. (2015)	*	*	/	*	*	*	*	*	6	
Xin et al. (2020)	*	*	*	*	**	*	*	*	8	
Tan et al. (2014)	*	*	*	*	**	*	*	*	8	
Qing et al. (2019)	*	*	*	*	**	*	*	*	8	
Notes:

* One point.

** Two points.

Results of meta-analysis

Gender

Three articles mentioned gender, heterogeneity test (I2% = 0%, P = 0.992), data analysis using a fixed-effects model, and meta-analysis results (Fig. 2) suggesting that gender is not a risk factor for recovery of motor nerve palsy after aneurysm surgery (OR = 0.75, 95% CI [0.51–1.10]).

Figure 2 Forest plot of the gender meta-analysis.

Age

Five articles mentioned age, heterogeneity test (I2% = 0%, P = 0.461), data analysis using a fixed-effects model, and meta-analysis results (Fig. 3) suggesting that age is not a risk factor for recovery of motor nerve palsy after aneurysm surgery (OR = 1.00, 95% CI [0.93–1.07]).

Figure 3 Forest plot of the age meta-analysis.

Aneurysm size

Five articles mentioned aneurysm size, heterogeneity test (I2% = 0.0%, P = 0.523), data analysis using a fixed-effects model, and meta-analysis results (Fig. 4) suggesting that aneurysm size is not a risk factor for recovery of motor nerve palsy after aneurysm surgery (OR = 0.85, 95% CI [−0.71 to 1.01]).

Figure 4 Forest plot of the aneurysm size meta-analysis.

Preoperative complete ONP

Seven articles mentioned preoperative complete ONP, heterogeneity test (I2% = 80.0%, P = 0.001), data analysis using a random-effects model, and meta-analysis results (Fig. 5) suggesting that preoperative complete ONP is a risk factor for recovery of motor nerve palsy after aneurysm surgery (OR = 2.27, 95% CI [1.07–4.81]). Because of the heterogeneity of the included studies, sensitivity analysis was performed using literature-by-exclusion, and the results of the analysis (Material S2, Fig. S1) suggested that the study was less sensitive, and the analysis was more stable.

Figure 5 Forest plot of the preoperative complete ONP meta-analysis.

Treatment time

Four articles mentioned treatment time, heterogeneity test (I2% = 65.4%, P = 0.034), data analysis using a random-effects model, and meta-analysis results (Fig. 6) suggesting that preoperative complete ONP is not a risk factor for recovery of motor nerve palsy after aneurysm surgery (OR = 1.01, 95% CI [0.91–1.13]). Because of the heterogeneity of the included studies, sensitivity analysis was performed using literature-by-exclusion, and the results of the analysis (Material S2, Fig. S2) suggested that the study was less sensitive, and the analysis was more stable.

Figure 6 Forest plot of the treatment time meta-analysis.

Surgery

Four articles mentioned surgery, heterogeneity test (I2% = 87.6%, P = 0.001), data analysis using a random-effects model, and meta-analysis results (Fig. 7) suggesting that surgery is a risk factor for recovery of motor nerve palsy after aneurysm surgery (OR = 9.88, 95% CI [2.53–38.57]). Because of the heterogeneity of the included studies, sensitivity analysis was performed using literature-by-exclusion, and the results of the analysis (Material S2, Fig. S3) suggested that the study was less sensitive, and the analysis was more stable.

Figure 7 Forest plot of the surgery meta-analysis.

Subarachnoid hemorrhage

Four articles mentioned subarachnoid hemorrhage, heterogeneity test (I2% = 13.4%, P = 0.326), data analysis using a fixed-effects model, and meta-analysis results (Fig. 8) suggesting that subarachnoid hemorrhage is a risk factor for recovery of motor nerve palsy after aneurysm surgery (OR = 1.29, 95% CI [1.06–1.56]).

Figure 8 Forest plot of the subarachnoid hemorrhage meta-analysis.

Publication bias

Publication bias was assessed using the egger test for risk factors and none were found to be present. gender (P = 0.603), age (P = 0.677), aneurysm size (P = 0.161), preoperative complete ONP (P = 0.121), treatment time (P = 0.488), surgery (P = 0.344), subarachnoid hemorrhage (P = 0.447).

Discussion

The present study is the first to use meta-analysis to assess the risk factors for poor recovery from oculomotor nerve palsy after aneurysm surgery, and it found that preoperative complete oculomotor nerve palsy, surgery, and subarachnoid hemorrhage were all risk factors for poor recovery after aneurysm surgery.

The pathophysiological mechanism of aneurysmal oculomotor nerve palsy is not well understood, and it was previously thought that the occupying effect of the aneurysm alone caused oculomotor nerve palsy (Lee, Hayman & Brazis, 2002). However, more and more reports have developed this theory, and it is now believed that the aneurysmal pulpability is also very important, and that it is not only the occupying effect that is the basis of the pathophysiology of motor nerve palsy (Hanse et al., 2008), but also there may be other pathogenetic mechanisms such as the nerve swelling caused by nerve migration or bruising due to aneurysm hemorrhage (Signorelli et al., 2020). Of course, there may be other pathogenetic mechanisms, such as nerve migration due to aneurysm hemorrhage or nerve swelling due to bruising, etc. We believe that the continuity of the axons and myelin sheath of the motor nerve exists in the early stage of the disease, and the conduction of action potentials is interrupted (White, Layton & Cloft, 2007). However, the conduction block is reversible, and as the aneurysm stimulates the nerve by beating and hammering, and as the duration of the stimulation increases, the degree of paralysis of the nerve increases (Wang, Kang & Wang, 2021), and damage to the axon and myelin sheaths occurs, reducing the number of neurons in the nerve, leading to an incomplete recovery (Kameda-Smith et al., 2022). These theories further support our conclusion that preoperative complete oculomotor nerve palsy will lead to a poor prognosis for patients after aneurysm surgery. In our study, subarachnoid hemorrhage was found to be an important factor affecting the recovery of moto neural function, but Chen et al. (2006) suggested that subarachnoid hemorrhage was not associated with the recovery of moto neural palsy. Kassis et al. (2010) and Mansour et al. (2007) suggested that subarachnoid hemorrhage is an important factor in the recovery of oculomotor nerve function, and Kassis et al. (2010) suggested that the mechanism of oculomotor nerve palsy due to rupture of posterior communicating artery aneurysm may be the effect of the clot on the oculomotor nerve rather than direct compression of the aneurysm, and that the absorption of the clot may facilitate recovery of ophthalmic muscle palsy. The results of the present study showed that the effect of the clot on the arterial nerve was not due to direct compression by the arteriovenous tumor, and that absorption of the clot facilitated the recovery of ophthalmoplegia (Seo et al., 2015). In the present study, the recovery of arterio-ocular nerve palsy was significantly better in patients with subarachnoid hemorrhage than in those without subarachnoid hemorrhage (Kheshaifati, Al-Otaibi & Alhejji, 2016). The better recovery of motor nerve palsy in patients with subarachnoid hemorrhage may be related to the rapid onset of the disease in patients with subarachnoid hemorrhage and the prompt diagnosis and treatment of patients with subarachnoid hemorrhage (Hou et al., 2022). The ONP associated with PcomAA may be due to direct mechanical compression of the aneurysm, which is relieved after surgical clamping and cannot be eliminated by endovascular embolization. Some studies have reported that the full recovery rate of ONP ranges from 32% to 85% (Yanaka et al., 2003), and with early surgical treatment, the full recovery rate can be as high as 88% (Stiebel-Kalish & Rappaport, 2007), whereas the full recovery rate after embolization ranges from 0% to 50% (Ahn et al., 2006), which further confirms our view that surgical procedures are able to improve the patient’s recovery. These findings suggest that clinicians need to consider the impact of surgery and patient’s specific condition on prognosis when evaluating and treating oculomotor nerve palsy. Future studies should delve into the mechanisms behind these factors, explore personalized treatment options, and conduct long-term follow-up studies to further improve patient recovery.

The current study still has several limitations: first, the number of included articles is small and most of them are from China, which may have a selection bias; second, the types of aneurysms in the included studies were inconsistent, which may have led to a large degree of heterogeneity in the articles; and third, the analysis process of the present study did not differentiate between OR, RR, or HR, and although there is not much difference in the actual difference among the three, there is a certain difference in the risk of disease among the three essentially measured diseases, which may lead to a certain degree of bias in the results.

Conclusion

Based on the results of the studies we included, we found that complete ONP before surgery led to poorer recovery, but patients with post-operative and subarachnoid hemorrhage had better recovery.

Supplemental Information

Supplemental Information 1 PRISMA checklist.

Supplemental Information 2 Specific retrieval strategies.

Supplemental Information 3 Rationale.

Supplemental Information 4 Sensitivity analysis.

Supplemental Information 5 Raw data.

Additional Information and Declarations

Competing Interests

Author Contributions

Data Availability

The authors declare that they have no competing interests.

Yuan Li conceived and designed the experiments, performed the experiments, analyzed the data, prepared figures and/or tables, authored or reviewed drafts of the article, and approved the final draft.

Ming Zhao conceived and designed the experiments, performed the experiments, analyzed the data, prepared figures and/or tables, and approved the final draft.

Xuemei Li conceived and designed the experiments, performed the experiments, analyzed the data, prepared figures and/or tables, and approved the final draft.

Tiejuan Liu conceived and designed the experiments, performed the experiments, analyzed the data, authored or reviewed drafts of the article, and approved the final draft.

Lin Zheng performed the experiments, analyzed the data, prepared figures and/or tables, authored or reviewed drafts of the article, and approved the final draft.

Deyu Hu performed the experiments, analyzed the data, prepared figures and/or tables, authored or reviewed drafts of the article, and approved the final draft.

Tongyan Liu performed the experiments, prepared figures and/or tables, authored or reviewed drafts of the article, and approved the final draft.

Lingyun Zhou performed the experiments, authored or reviewed drafts of the article, and approved the final draft.

The following information was supplied regarding data availability:

This is a systematic review/meta-analysis.

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
