# Peer review of "Risk factors for recovery from oculomotor nerve palsy after aneurysm surgery: a meta-analysis"

_PeerJ, doi:10.7717/peerj.18207_

## Round 0.1 · original submission · Minor Revisions

Both reviewers have only requested Minor Revisions. Please address all review comments and resubmit and in particular address the comments of R2 about the Conclusions

·

Basic reporting

The manuscript generally employs clear and professional English. However, there are several areas where the language could be improved for better readability and clarity. For example:

Line 12: "risk factors of recovery from oculomotor nerve palsy after aneurysm surgery" should be "risk factors for recovery from oculomotor nerve palsy after aneurysm surgery."
Line 25: "Patients with and subarachnoid hemorrhage will have better recovery" should be rephrased to "Patients with subarachnoid hemorrhage will have better recovery."

The manuscript provides a comprehensive background of oculomotor nerve palsy (ONP) and intracranial aneurysms (IA), citing relevant literature to support the context of the study. However, the introduction could benefit from a more detailed explanation of the knowledge gap the study aims to fill.

The article's structure conforms to standard scientific reporting with sections for the introduction, methods, results, and discussion. Figures and tables are relevant and appropriately labeled. The authors have included raw data, which is essential for transparency and reproducibility.

The manuscript is self-contained and presents results that are directly relevant to the stated hypotheses. The conclusions are supported by the data presented.

Experimental design

The study represents original primary research and aligns well with the aims and scope of the journal, focusing on risk factors for recovery from ONP after aneurysm surgery. The research question is clearly defined, addressing the recovery risk factors for ONP post-aneurysm surgery, which is a meaningful and relevant topic. However, the manuscript could benefit from a clearer statement of how it fills the existing knowledge gap in the introduction. The study appears to have been conducted rigorously, with appropriate ethical standards. The use of meta-analysis is suitable for the research question, and the methodology is sound. The methods are described in sufficient detail, allowing for replication of the study.

Validity of the findings

While the impact and novelty of the findings are not assessed in this review, the rationale and potential benefit to the literature are clearly stated, encouraging meaningful replication. The underlying data are provided, and the statistical analyses are robust and well-controlled. The use of multiple databases and a large sample size strengthens the validity of the findings. The conclusions are well articulated, directly linked to the original research question, and appropriately limited to the supporting results, providing a clear take-home message.

Additional comments

It would be helpful to include a more detailed discussion of the clinical implications of the findings and potential directions for future research.

·

Basic reporting

This article reports the meta-analysis performed to study the risk factors involved in the recovery of oculomotor nerve palsy due to aneurysm. The literature search has been done appropriately, statistical analysis including forest plots and heterogeneity tests are complete.

Experimental design

The research question is well defined and clinically relevant. The methodology is described well and information given is useful.

Validity of the findings

Rational of the study is described and all data have been provided. The conclusion needs to be modified as it does not support the results obtained.
Patients with surgery and subarachnoid hemorrhage have been mentioned as risk factors for recovery, but the conclusion states that they have better recovery.

Additional comments

Font size has to kept uniform. Capitalizing few words in the manuscript needs correction.

---

## Round 0.2 · accepted · Accept

Dear Authors,

Thank you for submitting your manuscript, titled "Risk factors for recovery from oculomotor nerve palsy after aneurysm surgery: a meta-analysis", to PeerJ. We are pleased to inform you that after careful consideration of the reviewers' comments and your revisions, your manuscript has been accepted for publication.

The reviewers highlighted several strengths in your work, including its relevance and originality as the first meta-analysis on this specific topic. They also praised the rigor of your methods, particularly your adherence to PRISMA guidelines, and commended the clarity of your research question. However, minor revisions were suggested regarding grammatical errors and phrasing. It is recommended that you ensure thorough proofreading by a native English speaker to address these points before final submission.

We appreciate your efforts in providing raw data, contributing to transparency and reproducibility in research. Congratulations again on the acceptance of your manuscript, and we look forward to seeing your final version for publication.

Best regards,

Dr. Pedrino

·

Basic reporting

The manuscript is generally written in clear, professional English. However, there are some grammatical errors and awkward phrasing that should be corrected (e.g. in the Abstract "risk factors of recovery from oculomotor nerve palsy after aneurysm surgery explored by meta-analysis."). I recommend careful proofreading and editing by a native English speaker.
Sufficient background is provided in the Introduction to contextualize the study. The research question and knowledge gap are clearly stated.
The structure generally conforms to standards, with defined sections. The tables and figures are relevant and well-described.
The authors have provided their raw data, which is commendable for transparency and reproducibility.

Experimental design

This is original primary research, a systematic review and meta-analysis, which falls within the scope of the journal.
The research question - identifying risk factors for recovery from oculomotor nerve palsy after aneurysm surgery - is well-defined, clinically relevant and addresses a stated gap in knowledge from prior individual studies.
The systematic review methods are rigorous and described in sufficient detail to replicate, including the literature search, inclusion/exclusion criteria, data extraction, and statistical analysis. The use of the PRISMA guidelines strengthens the methodology.

Validity of the findings

The authors have provided the underlying data for transparency. The meta-analysis methods and statistical approaches appear valid and robust.
The conclusions are succinct, clearly linked back to the original research question, and are appropriately limited to the supporting results without overinterpretation.
The discussion notes the novelty of this being the first meta-analysis on this specific topic. Meaningful conclusions are drawn regarding the key risk factors identified.